# Degraded Visibility Body-Specifically Affects Mental Rotation

**DOI:** 10.3390/bs14090784

**Published:** 2024-09-06

**Authors:** Zoé Rotach, Claude Beazley, Silvio Ionta

**Affiliations:** Sensory-Motor Lab (SeMoLa), Department of Ophthalmology-University of Lausanne, Jules Gonin Eye Hospital-Fondation Asile des Aveugles, 1004 Lausanne, Switzerland; zoe.rotach@unil.ch (Z.R.);

**Keywords:** visual impairment, mental imagery, body representation, sensorimotor, visuospatial

## Abstract

The way we perceive our own body is shaped by our perception. Changes in sensory input, such as visual degradation, can lead to visual-to-motor shifts in the reference frame used to mentally represent the body. While this effect has been demonstrated in mental representation of hands, it is still unknown whether it also affects mental representation of other body parts. To fill this gap, we asked 35 neurotypical participants to perform mental rotation (laterality judgement) of hand, foot, and full-body images, while the images’ visibility (figure/background contrast) was manipulated. Visibility deteriorations increased the steepness of the response time (RT) slopes for mental rotation of hand images shown from a less common view (palm) and of foot images from a more common view (dorsum), but not of full-body images from either the common or uncommon views. Suggesting that steeper and flatter RT slopes evoke the activation of a motor- or vision-based cognitive strategy for mental rotation, respectively, we propose that visual deterioration induces body-specific visual-to-motor shifts in mental processing. These findings show that the reliance on visual or motor aspects to mentally represent the body can be modulated by a reduction in sensory input, which changes the employed cognitive strategy.

## 1. Introduction

The largest part of our actions are based on mental imagery, which is a mental simulation ability that we use, consciously or unconsciously, to predict and eventually finetune our movements in response to internal goals and environmental requirements [1]. Without mental imagery, we would not be able to plan the different parameters (kinematics, force, etc.) of the movements required for touching a soap bubble versus grabbing a bowling ball.

In experimental settings, mental imagery can be investigated through mental rotation of body parts: participants are asked to judge the laterality (left, right) of an image representing a body part in different orientations, e.g., hands [2], feet [3], faces [4], and full bodies [5]. Typically, their performance in mental rotation is measured in terms of the time between the onset of the target image and the onset of the participant’s response (reaction time; RT). RT increases non-monotonically as a function of the orientation of the target image, with the shortest RT for the images presented at 0°, the longest RT for the images at 180°, and progressively decreasing RTs for further increases in orientation (180° to 360°), e.g., [6,7,8]. However, such an apparently straightforward relationship between RT and orientation is in fact modulated by many factors, including the type of image, the view from where the body part is shown, and the visibility of the image.

Regarding the type of image, mental rotation of hands, e.g., [9,10], faces [11,12], and feet [13,14] is typically associated with steeper RT slopes compared to mental rotation of full-body images, e.g., [15,16]. With respect to the view, mental rotation of body parts depicted from more familiar views (e.g., the dorsum of a hand) is more affected by image orientation than images shown from less familiar views (e.g., the palm of a hand) [17]. However, the relationship between view and orientation further depends on the type of image. Within-subject studies have shown that view strongly affects mental rotation of hand images, but it poorly influences mental rotation of full-body images [18]. Such a difference is corroborated by behavioral, e.g., [19] and brain imaging, e.g., [20] studies proposing that variations in RTs can be considered signs that different cognitive strategies are used to perform mental rotation. Steeper RT slopes would suggest that anatomical constraints are largely reflected in mental rotation (motor simulation strategy). Flatter RT slopes would suggest a weaker impact of the anatomical constraints (visual simulation strategy).

The variability of the interaction between the type of image, view, and orientation is particularly evident in neuropathological conditions. In particular, sensorimotor loss is associated with flatter RT slopes for tasks that would typically result in steeper RT slopes. It suggests that sensorimotor impairment could induce a shift from motor (steeper slopes) to visual strategies (flatter slopes) for mental rotation [21]. In fact, a backshift to motor strategies can be observed together with the sensorimotor benefits brought by restorative physiotherapy [14]. These shifts are in line with evidence from experimental settings. For instance, through instructions, it is possible to induce a switch between visuo-sensorimotor and purely visual strategies to perform mental rotation [22], and that the deterioration of visual inputs can trigger a visual-to-motor shift (opposite with respect to the one induced by spinal cord injury) in mental rotation [23]. In fact, degrading the visibility of the images (low figure/background contrast) triggers steeper orientation-related RT slopes for mental rotation of hand images shown from a view (palm) that would typically result in flatter RT slopes [23]. This suggests that visual deteriorations would be able to induce visual-to-motor shifts in the cognitive strategy to solve the task. However, this evidence is limited only to hand images, and therefore this conclusion risks underestimating the possible influence of mentally processing other body parts.

On this basis, we hypothesized that an interaction between image type (e.g., hand, foot, body), view (e.g., palm, dorsum), and visual degradation could affect the RT slopes of mental rotation of bodily images and therefore induce shifts in the use of motor or visual cognitive strategies. To test this hypothesis, we assessed the impact of deteriorated visual input on the RTs of mental rotation of hand, foot, and body images shown from the palm/plantar and dorsum views.

## 2. Methods

### 2.1. Participants

In line with previous work [23], thirty-five healthy participants (22 ± 2.4 years old) were enrolled in the experiment. Due to opposite findings about the influence of gender on mental rotation, which has been both supported [24,25] and excluded [26,27], we recruited both female (*N* = 27) and male (*N* = 8) participants. However, because hand dominance has been largely proven to affect mental rotation [2,28,29], all participants were right-handed according to the Edinburg Handedness Inventory [30].

### 2.2. Conditions and Stimuli

The stimuli comprised sets of gray-scale images of human bodily images (hand, foot, full body), presented in one out of four orientations with respect to the upright position (0°, 90°, 180°, 270°). The visibility of these images was manipulated (Low and High visibility). Keeping a fixed gray background, the Low visibility condition consisted of decreasing the contrast between the image and the background by 60% with respect to the High visibility condition (normal contrast; Figure 1). Hand images consisted in pictures of a male, adult, Caucasian human hand (Figure 1). In line with previous work [31] and based on their conceptual differences with respect to the hand, the other two classes of images (foot, body; adult, male, Caucasian) were selected as control stimuli.

The foot image represented a body part (like the hand) but a different one from the hand. Previous work showed that mental rotations of hand and foot images have different temporal dynamics [13,32,33,34,35]. In the present study, the comparison of mental rotation outputs for hand versus foot images provides important information about local body specificity of the effects of visibility manipulations.

Although the body images showed a hand like the hand images, the target hand was shown as part of a full body, not in isolation as in the hand images. Thus, the body images showed a front-facing person standing upright, bending the arms at the elbows with the hands at shoulder level, and with one hand darker than the other [19]. With respect to mental rotation of hand images, we selected this specific full body image based on its demonstrated ability to trigger different behavioral outcomes [18], activate different brain regions [20], and be sensitive to specific sensorimotor peculiarities [14,21]. The comparison of the results obtained with hand versus body images was based on previous work implementing whole-body images to investigate mental rotation, e.g., [5,16,36,37], clarifying the differences between local (hand) versus global (full body) mental processing.

All three types of images could show the dorsum or palm/plantar view of the relevant body part. In order to ease labeling, the plantar view of the foot images will be tagged as “palm” hereafter. The dorsum and palm views of each body stimulus had the same overall configuration and visual features (posture, gender, age, ethnicity, shape, size, etc.). All images could be either left- or right-lateralized. Left-lateralized images were mirror-reversed with respect to the right-lateralized ones. In sum, all images varied in terms of visibility, view, laterality, and orientation, and covered a visual angle of 10°–13° at 60 cm.

### 2.3. Procedure

Participants sat in front of a laptop positioned on a desk 60 cm from their eyes. They kept their hands on their laps, hidden from view under the desk. A microphone was positioned under the laptop [4]. The E-Prime 2 software (Psychology Software Tools Inc., Pittsburgh, PA, USA) controlled image presentation and recorded RTs. Before the experiment, all participants performed a training session with images presented at different orientations with respect to the experimental images. Each trial started with a fixation cross shown for 1 s. Then, an image appeared and remained on the screen until the participant gave a response. Participants had to classify each image as left- or right-lateralized, providing a verbal answer (“left” or “right”), as quickly and accurately as possible. Defined as the time between the onset of the image and the onset of participant’s verbal response, RT was considered the dependent variable of mental rotation task. For each trial, the image disappeared from the screen as soon as the microphone detected the first sound pronounced by the participant. Trial by trial, participants’ responses (left, right) were encoded manually by the experimenter. After this encoding, the following trial began. The experiment comprised six blocks, each concerning one visibility condition (High, Low) and one type of image (hand, foot, body), presented in both views (Palm, Dorsum), lateralities (left, right), and orientations (0°, 90°, 180°, 270°). The combination of visibility, view, laterality, and orientation determined 32 images for each type (hand, foot, body), each presented 3 times in each block, resulting in 96 images per block.

### 2.4. Data Analysis

Trials with incorrect responses or with RTs faster than 500 ms or slower than 3500 ms were filtered out from the following analysis [17,18,23,38,39,40] using the E-Prime 2 software (Psychology Software Tools Inc., Pittsburgh, PA, USA). The excluded trials corresponded to 8.2% of the total trials. After this preprocessing, all statistical analyses were performed with the STATISTICA 12 software (StatSoft Inc., Tulsa, OK, USA). RT data were analyzed by a four-way repeated-measure ANOVA with Stimulus (Hand, Foot, Body), Visibility (High, Low), View (Palm, Dorsum), and Rotation (no, medial, upside down, lateral) as main factors. While lateral rotations (Lat) included right hands (Dorsum and Palm views) presented at 90° and left hands (Dorsum and Palm views) presented at 270°, medial rotations (Med) included right hands (Dorsum and Palm views) presented at 270° and left hands (Dorsum and Palm) presented at 90° [23,41,42,43]. This data classification avoids any potential bias due to image laterality, is sensitive to visibility manipulations [23], and is based on previous evidence that mental rotation is faster for images oriented towards than away from the participant’s midsagittal plane [41,44,45]. Images presented with fingers/toes pointing upright or downwards were classified as No or Upside Down (UpDn) rotations, respectively. Thus, each mental rotation trial was classified in one out of four rotations (No, Med, UpDn, Lat). The partial eta-squared (η^2^p) established the effect size of the significant main effects and interactions, with a confidence interval (CI) at 90%, and the lower (CI_low_) and upper (CI_high_) limits of the CI computed for each η^2^p. Bonferroni corrections for multiple comparisons were applied to the post hoc tests for the significant main effects and interactions.

## 3. Results

### 3.1. Visibility Effects

The ANOVA on RTs showed the significant main effect of Visibility [F(1,33) = 30.2; *p* < 0.001; η^2^p = 0.48; CI_low_ = 0.25; CI_high_ = 0.61] and the four-way interaction between Stimulus, Visibility, View, and Rotation [F(6,198) = 2.4; *p* = 0.026; η^2^p = 0.07; CI_low_ = 0.004; CI_high_ = 0.1]. The main effect of Visibility was explained by the longer RTs for Low visibility (1680.9 ms) compared to High visibility (1488.5 ms). The post hoc comparisons of the four-way interaction between Stimulus, Visibility, View, and Rotation showed that, compared to High visibility, Low visibility was associated with steeper RT slopes for the Hand/Palm images but not the Hand/Dorsum images (Figure 2), for the Foot/Dorsum images but not the Foot/Palm images (Figure 3), and not for either the Body/Palm or Body/Dorsum images (Figure 4).

In particular, the post hoc comparisons showed that mental rotation of Hand/Palm images was significantly affected by Rotation in Low visibility, in that the RT for the UpDn rotations (2099.6 ms) were significantly longer than the No (1649.6 ms; *p* < 0.001) and Lat rotations (1760.2 ms; *p* = 0.006). In High visibility, this pattern was absent [the RTs for all rotations were not statistically different from each other (No = 1645.9 ms; Med = 1842.4 ms; UpDn = 1750.2 ms; Lat = 1640.9 ms)]. Mental rotation of Hand/Dorsum images was not particularly affected by Visibility, in that the RT slopes for Low and High visibility were very similar. In High visibility, the RTs for the UpDn rotations were significantly the longest (High = 1978.6 ms) with respect to the other rotations (No = 1510.5 ms, Med = 1449 ms, Lat = 1538.9 ms). In Low visibility, the RT for UpDn rotation (2045.2 ms) was significantly longer with respect to No (1632.4 ms; *p* < 0.001) and Med rotation (1629.2 ms; *p* < 0.001) (Figure 2).

For the Foot images, the pattern was the opposite. In fact, Visibility affected mental rotation of Foot/Dorsum images but not the Foot/Palm images. In particular, while mental rotation of Foot/Dorsum images was significantly affected by Rotation in Low visibility [RTs for the UpDn rotations (1910.2 ms) were significantly longer than the No (1462.8 ms), Med (1495.6 ms), and Lat rotations (1590.8 ms) (all *p*_s_ < 0.02)], this pattern was absent in High visibility (No = 1217.3 ms; Med = 1295.4 ms; UpDn = 1519.7 ms; Lat = 1348.4 ms). Conversely, mental rotation of Foot/Palm images was not affected by Rotation in either Low (No = 1804.8 ms; Med = 1897.4 ms; UpDn = 1875.5 ms; Lat = 1934.8 ms) or High visibility (No = 1687 ms; Med = 1711.6 ms; UpDn = 1777.5 ms; Lat = 1721.3 ms) (Figure 3).

For the Body images, Visibility did not significantly affect mental rotation of either the Palm-view (Low visibility: No = 1436.4 ms, Med = 1521.9 ms, UpDn = 1650.4 ms, Lat = 1524.6 ms; High visibility: No = 1255.7 ms, Med = 1248.4 ms, UpDn = 1356.2 ms, Lat = 1250.3 ms) or Dorsum-view images (Low visibility: No = 1357.1 ms, Med = 1423.9 ms, UpDn = 1558.6, Lat = 1422.7 ms; High visibility: No = 1237.7 ms, Med = 1226.6 ms, UpDn = 1308.2 ms, Lat = 1205.3 ms) (Figure 4).

### 3.2. Other Effects

The other significant effects generally confirmed previous findings, including the main effects of Stimulus [F(2,66) = 10.8; *p* < 0.001; η^2^p = 0.05; CI_low_ = 0; CI_high_ = 0.14], View [F(1,33) = 38.1; *p* < 0.001; η^2^p = 0.53; CI_low_ = 0.32; CI_high_ = 0.65], and Rotation [F(3,99) = 25.6; *p* < 0.001; η^2^p = 0.43; CI_low_ = 0.3; CI_high_ = 0.52]. The main effect of Stimulus was explained by the shorter RTs for Body (1374 ms) with respect to Hand (1739.5 ms) and Foot images (1640.6 ms) (all *p*_s_ < 0.0051). The main effect of View was due to the longer RTs for images seen from the Palm view (1664.8 ms) compared to the Dorsum view (1504.7 ms). The main effect of Rotation was due to the longer RTs for UpDn images (1735.8 ms) with respect to all other rotations (No = 1491.4 ms, Med = 1554.4 ms; Lat = 1557.1 ms; all p_s_ < 0.001), as well as to the significant difference between the RTs for the No and Med rotations (*p* = 0.035). The significant interaction between Stimulus and View [F(2,66) = 8.7; *p* < 0.001; η^2^p = 0.2; CI_low_ = 0.07; CI_high_ = 0.32] showed that the difference between the Palm- and Dorsum-view images was significant for the Foot (Palm = 1801.2 ms; Dorsum = 1480 ms; *p* < 0.001) but not for the Hand (Palm = 1787.5 ms; Dorsum = 1691.4 ms) or the Body images (Palm = 1405.5 ms; Dorsum = 1342.5 ms). The interaction between View and Rotation [F(3,99) = 4.8; *p* = 0.004; η^2^p = 0.12; CI_low_ = 0.03; CI_high_ = 0.21] indicated that RTs for Low visibility (No = 1579.9 ms, Med = 1688.9 ms, Lat = 1638.7 ms) were significantly longer than in High visibility (No = 1402.9 ms, Med = 1419.9 ms, Lat = 1475.6 ms) in all the rotations except UpDn (Low = 1751.6 ms, High = 1720.1 ms). The interaction between Stimulus, View, and Rotation [F(6,198) = 2.8; *p* = 0.011; η^2^p = 0.08; CI_low_ = 0.009; CI_high_ = 0.1] revealed that, overall, View affected mental rotation of Hand and Foot but not Body images; for the Hand and Foot images (but not Body images), mental rotation of Palm-view images was generally slower than the Dorsum-view images. This interaction is further explained by the four-way interaction between Stimulus, Visibility, View, and Rotation.

## 4. Discussion

Mental rotation is tightly linked to visual processing [46], visual complexity [47], and visual deterioration [23]. Variations in RT slopes shed light on the type of cognitive strategy used to complete experimental tasks, in that steeper slopes would hint to motor strategies, while flatter slopes would be signs of visual strategies [14,18,20,21,23]. The present study extends this evidence, revealing the specificity of visual-to-motor shifts in the cognitive strategy used to perform mental rotation. We show that the effects induced by degrading the visual input are specific for mental representation of body parts that in typical conditions rely mostly, but not exclusively, on visual processing. In fact, for hand images, the visual-to-motor effects of Visibility were evident in the palm-view images, whose mental rotation typically relies mostly on visual processing (flat slopes). Visibility did not affect the dorsum-view hand images, where motor strategies would already play a major role. Conversely, mental rotation of foot images was affected by visibility manipulations only when the dorsum-view (not plantar-view) images were shown. Finally, mental rotation of body images was not affected by Visibility in either the palm- or dorsum-view images. These visual-to-motor shifts are in line with the clinical observation that visual [48] or motor impairments [49] affect the confidence on the deteriorated system and increase the trust in alternative more reliable sensory input.

### 4.1. Body-Specific Visual-to-Motor Shifts

#### 4.1.1. Stimuli-Related Effects

In line with previous evidence [23], we found that decreased Visibility (Low vs. High) affected mental rotation of Hand images seen from a relatively less familiar view (Palm) but not from a more familiar one (Dorsum). The relatively flatter RT slope for the Palm-view Hand images in High visibility suggests that, in typical conditions, mental rotation of these images is based mostly on visual strategies. Low visibility seemed to change this pattern, with the RT slope being steeper, implying that decreased visibility induced a shift to a more motor-based strategy for mental rotation. Such a shift was not present for Dorsum-view Hand images, as the RT slopes for mental rotation of High- and Low-visibility dorsum-view images were similarly steep. The independence of mental rotation of Dorsum-view Hand images from visibility manipulations suggests that a motor strategy was already in place for this task. Therefore, the additional motor load eventually brought by visual deteriorations did not affect or overtake the motor-biased plateau for mental rotation of Dorsum-view Hand images. In other words, mental rotation of Dorsum-view Hand images would be already so strongly based on motor cognition that the visual deteriorations of the present experiment would not be able to further increase the reliance on motor strategies. This interpretation is further corroborated by the fact that Low visibility reduced the difference between the RT slopes of Dorsum- and Palm-view Hand images but did not affect the RT slopes of mental rotation of Dorsum-view Hand images.

Mental rotation of both Palm- and Dorsum-view Foot images was not affected by Visibility. The RT slopes for Palm-High and Palm-Low images were not significantly different. This was in contrast with the corresponding comparison for the Hand images, where the RT slope for Palm-Low Foot images was steeper with respect to that for Palm-High Foot images. Mental rotation of Dorsum-view Foot images was affected by Visibility as follows: the RT slope for Dorsum-Low Foot images was steeper than Dorsum-High Foot images. Altogether, these findings suggest that mental rotation of Palm-view Foot images would be so profoundly based on a visual strategy (flat RT slope) that the effects of degrading the visual input brought by the present study would not be strong enough to trigger a shift to a motor strategy. On the other hand, considering that, by default, motor mechanisms may play a larger role in mental rotation of Dorsum- than Palm-view Foot images, it seems possible that the effects of visual deterioration would sum up to these default motor components. Therefore, they would render the visual-to-motor shift more noticeable in Dorsum- than Palm-view Foot images. This interpretation is supported by the fact that the RT slopes for Dorsum/High Foot images appeared steeper than for Palm/High Foot images.

Visibility did not affect mental rotation of Body images from either the Dorsum- or Palm views. The RT slopes for High- and Low-visibility Body images matched for view were similar, indicating that Visibility did not affect the performance.

#### 4.1.2. Palm-View-Related Inter-Stimulus Comparisons

The typically flatter RT slopes for Hand images (Palm-High) became steeper in Low visibility. The effects of degraded visibility for Palm view were specific for Hand images, as they were absent in mental rotation of both Foot and Body images. Typically, the RT for mentally rotating palm-view images of hand, foot, and body is minimally affected by the orientation of the image [18,33]. This aligns with the flatter RT slope that we found for mental rotation of all palm-view images in High visibility. Degraded visibility made the RT slopes steeper for the Hand but not the Foot and Body images. We interpret this finding as a sign that mental rotation of Palm-view Hand images is based on visual processing, but also contains some motor aspects that are closer to be activated by contextual factors (degraded visibility) with respect to Foot and Body images (which would be more strongly based on visual strategy). This implies that degraded visibility, or at least the amount of visual degradation employed in the present study, can prompt a visual-to-motor shift in the cognitive strategy employed to process mental representation of a body part that already contains relatively more motor features (hand). This fine-tuned adjustment based on visibility underscores the recalibration of mental body representations in response to the available visual information. This is consistent with previous findings suggesting that mental rotation of hands is also sensitive to stimulus orientation in blind individuals [50,51]. It has been recently proposed that a significant influence of rotation indicates the activation of a motor strategy for mental rotation, whereas a lesser impact of rotation supports the utilization of a visual strategy [23]. The present results demonstrate that mental rotation of Palm-view images was based more on motor processing for Hand than Foot and Body images. This implies that decreased visibility prompted the adoption of a motor strategy for mentally rotating a type of image (Palm/Hand) that is typically rotated using a visuospatial strategy but still has more motor properties with respect to other stimuli (Palm/Foot, Palm/Body).

#### 4.1.3. Dorsum-View-Related Inter-Stimulus Comparisons

Mental rotation of Dorsum-view Hand images was mostly independent from Visibility, in that the RT slopes were similarly steep in Low and High visibility. This lack of impact is likely because Dorsum Hand images are typically mentally processed using motor strategies as the primary frame of reference, as evidenced by steep RT slopes for Dorsum-High images [43]. Therefore, the eventual motor-leaning effect of decreased visibility would sum up to an already existing motor-based mental processing for Hand images, resulting in unnoticeable differences. However, Dorsum-view images were not indiscriminately independent from visual deteriorations. Indeed, mental rotation of Dorsum-view Foot images was affected by Visibility, in that the RT slope was steeper in Low than High visibility. In this context, it is likely that the amount of sensory disturbance brought by the visibility degradation of the present study was sufficient to render the visual reference frame less reliable. This would facilitate a visual-to-motor shift based on the pre-existent motor aspects already in place for mental rotation of Dorsum-view Foot images. This process would not occur for the Hand images because these images would already be strongly based on motor processing, nor in the Body images because they would be too strongly based on visuospatial processing (flat RT slopes). In fact, mental rotation of Body images was not affected by Visibility, with the RT slopes being similarly flat in both Low and High visibility. In sum, the effects induced by the visual degradation of the present study would affect mental rotations that already contain some, but not only, motor components (Foot), while they would not be strong enough to affect mental rotations already based primarily on motor (Hand) or visual (Body) strategies.

### 4.2. Visual Perturbation and Body Representation

Our interpretations in terms of visual-to-motor shifts are connected to the differentiation between the psychological concepts of body schema and body image [52]. Body schema refers to the sensorimotor aspects of the body [53], mental framework used to control body movements [54], and the internal bodily sensorimotor states [55,56]. Body image concerns the visual aspects of the body [57], drawing upon prior visual experiences [58,59], and involves an external viewpoint of one’s own body [60]. Under normal circumstances, body schema and body image are seamlessly integrated, but certain clinical conditions (e.g., sensory deprivation) can profoundly impact their mutual interaction [61]. Thus, a reduction in sensory input, such as somatosensory loss [21] or visual perturbation [23], can disrupt the relationship between body schema and body image. In this vein, the present study shows that unreliable visual input (decreased visibility) can induce a visual-to-motor shift from body image to body schema in mental processing of images that in normal conditions would rely on visuospatial processing (Palm-view images). The concept of the body image-to-schema shift due to visual deteriorations aligns with previous evidence that blindness can prompt the adoption of alternative strategies for mental rotation [62] and is associated with abnormal patterns of brain activity during mental rotation [63]. The utilization of the body schema as a more dependable reference frame in cases of low vision corresponds to the notion of functional changes following or accompanying vision loss or deterioration. Indeed, there is ample evidence demonstrating that visually impaired individuals employ alternative strategies to compensate for the lack of visual input, including a greater reliance on somatosensory information [64]. Such perceptual adaptations may drive or support the cognitive [65] and behavioral [66,67] adjustments necessary to achieve accuracy levels comparable to typical visual conditions [68].

### 4.3. Limitations and Future Perspectives

The present study included both female and male participants. It might be argued that the results could be affected by potential gender/sex-related differences in mental rotation skills. However, the relationship between gender/sex and mental rotation can be controversial, in that it has been supported by some studies [24,25] but excluded by others [26,27]. In addition, rather than by gender/sex as a whole, mental rotation seems to be more influenced by variations in sex hormones like testosterone, estradiol, and progesterone [69,70,71,72]. Investigating this topic could be the focus of future studies, which could recruit female participants during, for instance, early and late follicular phases, when estradiol and progesterone levels are different. Similarly, participants using hormone-based oral contraceptives could be recruited during the inactive pill phase. This approach would help to recruit homogeneous groups of participants in terms of hormonal profiles and could provide more precise insights into the influence of gender/sex versus hormones on mental rotation.

## 5. Conclusions

This study shows that reducing visual input led to a visual-to-motor shift in the reference frame for mentally representing and manipulating the specific body parts typically associated with visuospatial reasoning (Palm-view hands). These findings suggest that body-related visual input can alter body representation, prompting a transition from visual to motor strategies. The study’s significance is twofold. Firstly, it provides a direct and within-subject comparison of how visual deteriorations affect mental representation of the body. Secondly, it underscores the importance of visual input, indicating a heightened reliance on either motor or visual aspects in mental representation of the body. This highlights the existence of vision-dependent mechanisms which are particularly valuable in clinical settings, where a decline or improvement in visual abilities may coincide with changes in body representation. Understanding these mechanisms could have important implications for populations affected by visual deficits, as well as for training and rehabilitation programs.

## Figures and Tables

**Figure 1 behavsci-14-00784-f001:**
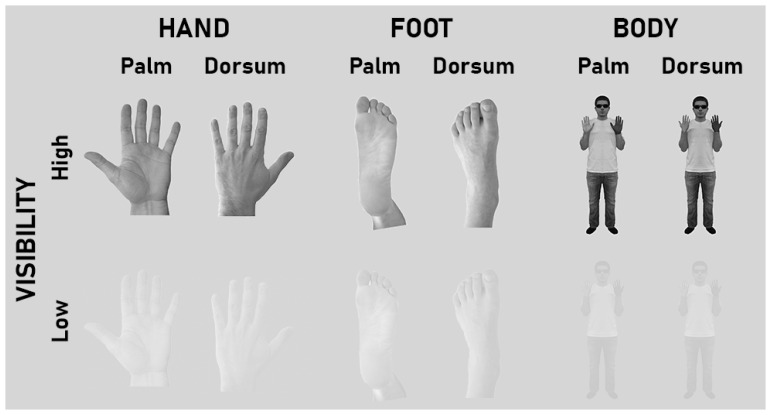
Experimental stimuli. All images (Hand, Foot, Body) were presented in both typical (High visibility) and degraded (Low visibility) visual conditions. In Low visibility, the contrast of the images was 60% lower with respect to High visibility. The background (gray) was the same for High and Low visibility. To ease inter-image comparisons, the plantar view of the foot images was tagged as “Palm”. All images showed bodily images from either the Palm or Dorsum views, were rotated in one out of four orientations (0°, 90°, 180°, 270°), and varied in terms of laterality (left, right). For illustrational purposes, the figure shows only left-lateralized images at 0°.

**Figure 2 behavsci-14-00784-f002:**
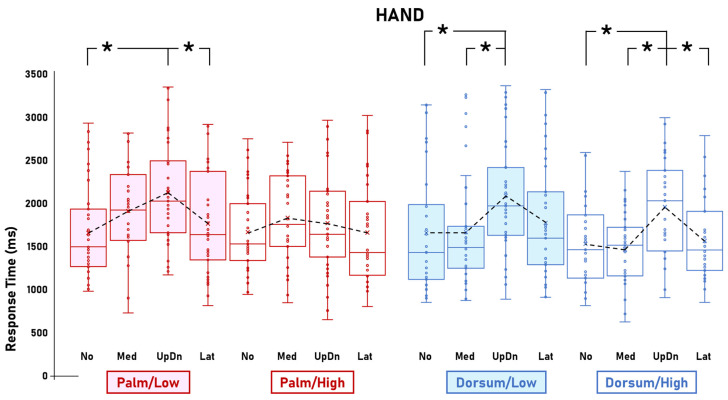
Hand. Response times for mental rotations of Hand images as a function of Visibility (Low, High), View (Palm, Dorsum), and Rotation (No, Med, UpDn, Lat). Visibility manipulations affected mental rotation of Palm-view images, in that Low visibility was associated with a stronger influence of Rotation with respect to High visibility. This effect of Visibility was absent in mental rotation of Dorsum-view images. The figure shows the Hand-related data of the significant interaction between Stimulus, Visibility, View, and Rotation. Dots represent datasets. Crosses represent averages. Black dashed lines represent RT slopes. Asterisks represent statistically significant differences.

**Figure 3 behavsci-14-00784-f003:**
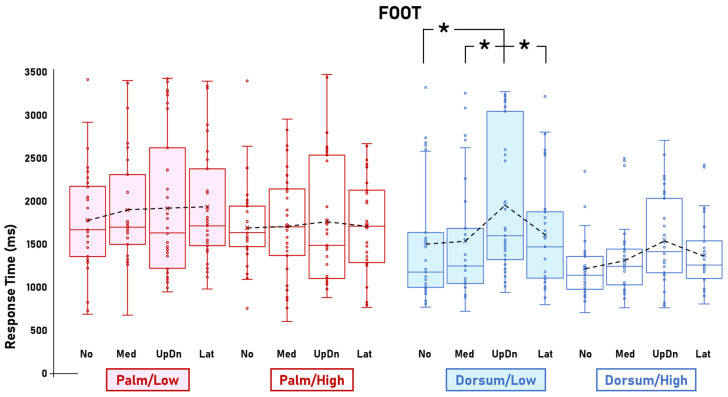
Foot. Response times for mental rotations of Foot images as a function of Visibility (Low, High), View (Palm, Dorsum), and Rotation (No, Med, UpDn, Lat). Visibility manipulations affected mental rotation of Dorsum-view images, in that Low visibility was associated with a stronger influence of Rotation with respect to High visibility. This effect of Visibility was absent in mental rotation of Palm-view images. The figure shows the Foot-related data of the significant interaction between Stimulus, Visibility, View, and Rotation. Dots represent datasets. Crosses represent averages. Black dashed lines represent RT slopes. Asterisks represent statistically significant differences.

**Figure 4 behavsci-14-00784-f004:**
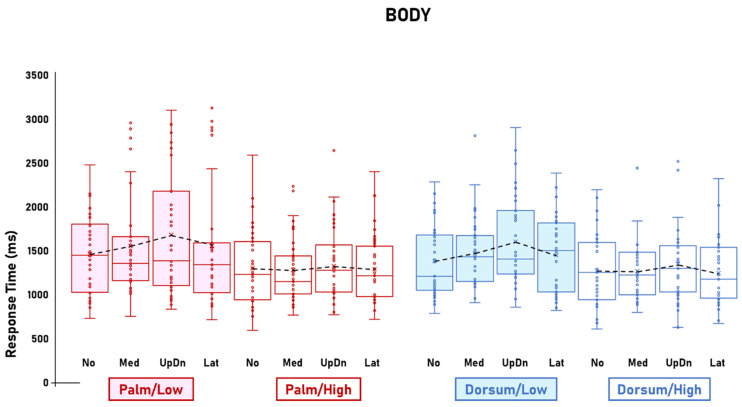
Body. Response times for mental rotations of Body images as a function of Visibility (Low, High), View (Palm, Dorsum), and Rotation (No, Med, UpDn, Lat). Visibility manipulations did not affect mental rotation of either Dorsum- or Palm-view images. The figure shows the Body-related data of the significant interaction between Stimulus, Visibility, View, and Rotation. Dots represent datasets. Crosses represent averages. Black dashed lines represent RT slopes.

## Data Availability

The data presented in the present study are openly available in Zenodo at https://zenodo.org/records/13219514 (accessed on 4 August 2024).

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
