# Peer review of "Degraded Visibility Body-Specifically Affects Mental Rotation"

_behavsci, 2024, doi:10.3390/bs14090784_

Round 1

Reviewer 1 Report

Comments and Suggestions for Authors

Congratulations for a manuscript very solid, with great interest and convenient justification.

Just a few questions arise after my reading. 

It is not very clear for the reader (at least, in my case) what the outcome or dependent variable is. When you read it, it is unclear how the mental rotations finally derive in a given response. I have finally found that the response required is verbal: right/left, but maybe, a subsection of variables is needed, only to clarify.

To me, it is not obvious why responses above 3500 ms should be canceled. I understand that very fast responses could be caused by impulsive tendencies, but I ignore the reason about cancelling responses longer than 3.5 s, which is not excessive, from my point of view.

In the Discussion, when it is defended how a visual strategy is used in mental rotation, in comparison to motor strategies, made me think about the difficulty to use motor strategies in your study, because of the shorter time for response analyzed here. Perhaps you have more data, but it seems rare to discuss whether your sample relied more on visual strategies when a period of 3.5 s to answer avoids many motor attempts. 

At the same time, I am not too sure that a palm view is less familiar in general. You are the expert, but I would think that palm and dorsum, when speaking about hands, have similar frequencies of being viewed.

Very interesting the discussion and justification of how to move from the body image to the body schema.

Certain references do not contain the doi number: Wijntjes et al., 2008, Vinter et al., 2012, etc... but others do.

Reviewer 2 Report

Comments and Suggestions for Authors

“Degraded visibility body-specifically affects mental rotation”( behavsci-3164628)

Visual degradation can lead to visual-to-motor shifts in the reference frame used to mentally represent the body, such as hand. Whether this effect can be transferred to other body parts, such as foot, and full-body was investigated by using 3 Stimulus (Hand, Foot, Body)×2Visibility (High, Low)×2 View (Palm, Dorsum)×4 Rotation (no, medial, upside-down, lateral) design in the current investigation. The results revealed that visibility deteriorations increased the steepness of the response time (RT) slopes for the mental rotation of hand images shown from less common views (palm), of foot images from more common view (dorsum), but not of full-body images from either common or uncommon views. These results suggest that the reliance on visual or motor aspects to mentally represent the body can be modulated by a reduction in sensory input, changing the employed cognitive strategy. Overall, this topic is interesting and the findings hold great practical implications. The manuscript is overall well presented. However, some concerns appeared after reading the whole manuscript.

1. How did you determine the sample size? Did you calculate the sample size needed before formal study?

2. Are you sure influence of gender on mental rotation is uncertain”? please refer to literature review about the gender differences on mental rotation in the following paper, especially the meta-analysis and reviews mentioned in the following paper.

Zhang, K., Fang, H., Li, Z., Ren, T., Li, B. M., & Wang, C. (2024). Sex differences in large-scale brain network connectivity for mental rotation performance. NeuroImage, 120807.

It would be interesting to see the results if you exclude the data from male participants.

3. A separate part about the limitations and future directions would be inspiring if added in the discussion part.

4. The current introduction is not friendly to the readers since it is only included one long paragraph. Please separate it into paragraphs to ease the readers’ burden and make the flow of introduction part follow a more logic way.

5. Please revised all the figures about the indication of the significance of difference, it is not clear which two conditions have significant difference from the current three figures.

6. For the Experimental stimuli, it had better to provide all the Experimental stimuli as supplementary material for replications in the future.

7. The abbreviation of confidence interval should be CI.

8. “4.3. Conclusions” should be 5. Conclusions.

Comments on the Quality of English Language

Minor editing of English language required.

Round 2

Reviewer 2 Report

Comments and Suggestions for Authors

Thanks for the revisions and no further concerns.